# Observation of GHG vertical profile in the boundary layer of the Mount Qomolangma region using a multirotor UAV

Ying Zhou[1,2], Congcong Qiao[1,2], Minqiang Zhou[1,2], Yilong Wang[2,3], Xiangjun Tian[2,3], Yinghong Wang[4], Minzheng Duan[1,2], and Yilong Wang[2,3]

[1]Key Laboratory of Middle Atmosphere and Global Environment Observation, Institute of Atmospheric Physics, Chinese Academy of Sciences, Beijing 10029, China
[2]University of Chinese Academy of Sciences, Beijing 100049, China
[3]State Key Laboratory of Tibetan Plateau Earth System, Environment and Resources, Institute of Tibetan Plateau Research, Chinese Academy of Sciences, Beijing 100101, China
[4]Public Technology Center, Institute of Atmospheric Physics, Chinese Academy of Sciences, Beijing 10029, China

**Correspondence:** Minzheng Duan (dmz@mail.iap.ac.cn) and Yilong Wang (wangyilong@itpcas.ac.cn)

**Abstract.** Understanding the vertical profile of greenhouse gases (GHGs) is crucial for elucidating their sources and sinks, transport pathways, and influence on Earth's radiative balance, as well as for enhancing predictive capabilities for climate change. Remote sensing methods for measuring vertical GHG profiles often involve substantial uncertainties, while in-situ measurements are limited by high equipment costs and operational expenses, rendering them impractical for long-term contin-
uous observation efforts. In this study, we have developed an automatic low-cost and user-friendly multi-altitude atmospheric sampling device designed for small and medium-sized unmanned aerial vehicles (UAVs), balloons, and other flight platforms. A field campaign was carried out in the Mount Qomolangma region, at an average surface altitude of 4300 m above sea level (a.s.l.). In total, we conducted 15 flights with 139 samples from the ground surface up to a height of 1215 m using the device mounted on a hexacopter UAV platform. The samples were analyzed using the Agilent gas chromatography (GC) 7890A, and
the vertical profiles of four GHG species ($CO_2$, $CH_4$, $N_2O$, and $SF_6$) were archived. The new data depict the vertical distribution of GHGs in the boundary layer of the Mount Qomolangma region. To apply this method for long-term monitoring of small UAVs, future efforts must focus on reducing the weight of the equipment and improving the sampling efficiency.

## 1  Introduction

Contemporary global warming, primarily driven by human activities, is an urgent environmental challenge marked by an
increase in the atmospheric concentration of greenhouse gases (GHGs), causing a rapid rise in global temperature since the Industrial Revolution(Masson-Delmotte et al., 2019; Friedlingstein et al., 2023). Monitoring the changes in GHG concentration is essential for understanding climate change and promoting environmental protection. Carbon dioxide ($CO_2$) is the most potent GHG, whose radiative forcing has reached $1.82 \pm 0.19 W/m^2$ from 1750 to 2019 (on Climate Change , IPCC), followed by methane ($CH_4$), nitrous oxide ($N_2O$) and other GHGs. The concentrations of GHGs are influenced by surface fluxes and
atmospheric chemical transport, leading to spatial distributions that are not uniform. Variations in emissions from natural and

anthropogenic sources and atmospheric circulation patterns result in significant differences in greenhouse gas concentrations at different altitudes(Carnell and Senior, 1998; Ren et al., 2011; Xie et al., 2013).

Vertical distribution of these GHGs is useful to elucidate their sources and sinks, as well as the vertical mixing of the atmosphere. For instance, the vertical profiles of $CO_2$ observed by aircraft were used for diagnosing errors in the simulation of surface $CO_2$ fluxes (Jin et al., 2024) and have been integrated into inverse modeling of carbon fluxes (Niwa et al., 2012; Jiang et al., 2013). It has been shown that satellite-based $CO_2$ inversions can yield results comparable to surface network inversions when supplemented with aircraft observations to reduce errors(Chevallier et al., 2019). Furthermore, combining surface-based and space-based $CO_2$ measurements within a flux inversion framework improves constraints on regional $CO_2$ fluxes compared to using either data source alone(Byrne et al., 2020). Additionally, the vertical distribution of GHGs provides key prior values for satellite remote sensing retrieval algorithms, enhancing the accuracy of satellite retrievals (Ramanathan et al., 2018; Zhongxiu BAO and YAO, 2020). The Global Carbon Project (GCP) also recommends this type of data (Friedlingstein et al., 2022).

There are two primary methods for obtaining the vertical distribution of atmospheric GHGs: indirect measurements (remote sensing technique) and direct measurements. The first approach involves analyzing the observed characteristic spectrum through space-based satellites or payloads (Buchwitz et al., 2005; O'Dell et al., 2012; Yoshida et al., 2013), ground-based Lidar (Kuma et al., 2021), and high-resolution spectrometers (Wunch et al., 2011). The accuracy of indirect measurement methods for quantifying GHG sources and sinks is limited by several factors, such as cloud cover, aerosols, and surface reflections. These limitations lead to considerable uncertainty and limited spatial resolution of GHG data, as well as challenges in detecting localized changes in GHG sources and sinks.

The direct measurement technique requires the use of specialized equipment capable of accurately measuring the atmosphere's composition, such as Cavity Ring-down Spectroscopy (CRDS) (Wheeler et al., 1998; Wilkinson et al., 2018). To acquire vertical distribution information, multiple inlets are often installed at different altitudes of a tower, which typically only extends a few hundred meters (Haszpra et al., 2012). Alternatively, measurements can be carried aloft by planes (Sun et al., 2020) or balloons (Li et al., 2014; Zhongxiu BAO and YAO, 2020), although these methods come with significant logistical, cost, and airspace limitations. Compared to remote sensing, direct measurements provide higher precision and vertical resolution for GHG data that can be easily tied to the calibration standards. Recently, advancements in Unmanned Aerial Vehicles (UAVs) have provided a lightweight, easy-to-operate, and easily recoverable platform for vertical observations. Due to their small size, portability, and low cost, UAVs have emerged as a popular method for obtaining the distribution of atmospheric constituents, effectively overcoming the limitations of traditional methods (Glaser et al., 2003; Neumann and Bartholmai, 2015; Etts et al., 2015; Brosy et al., 2017; Chang et al., 2020).

Many works have used UAVs for in-situ observation of GHGs, primarily utilizing Non-Dispersive Infrared (NDIR) sensors to measure $CO_2$ and $CH_4$ (Kunz et al., 2018; Reuter et al., 2021; Britto Hupsel de Azevedo et al., 2022; Han et al., 2024). While NDIR and other low-cost sensors have the advantage of real-time and continuous monitoring due to their lightweight design, they face challenges such as frequent calibration and potential fluctuation due to the change of ambient environments such as pressure, temperature and also vapor content in the variable atmosphere along with altitude (Liu et al., 2022). In contrast,

flask(usually made of metal) analysis allows air samples to be collected and analyzed in a controlled laboratory (Loftfield et al., 1997), but flask evacuation and cleaning is more labor-intensive, and it is not easy to operate in-flight measurements. We have developed a device similar to flask sampling but aluminum bags are used, featuring a lighter design, and expanded its capabilities to analyze additional GHG components. Note that our system requires a higher payload capacity and larger platform size than online analysis sensors. This portable device operates automatically and can collect air samples from multiple altitudes in a short period. Comprehensive indoor tests verified the device's sampling speed and liability for field measurements. The device was used in a five-day campaign of field measurements on Mount Cho Oyu Basecamp (4950 m a.s.l.) and Mount Qomolangma Station (4300 m a.s.l) between 29 September and 03 October 2023. The device was taken by a medium-sized UAV up to 1250 meters above the ground. During the flights, air samples were collected at different altitudes form the ground to the upper air. The samples were then analysed by a chromatography to derive gas concentrations, including $CO_2$, $CH_4$, $N_2O$, and $SF_6$.

The paper is structured as follows: Section 2 provides an overview of the gas collection system, and outlines the sampling and analysis procedures used in this experiment. Section 3 details the field experiments, including site descriptions and a discussion of the results. Finally, Section 4 summarizes the key findings and their implications.

## 2 Methodology

### 2.1 Gas collection system

The schematic of the automatic sampling device is shown in Figure 1. Airbags are used to collect air samples. Each airbag is a vacuum-sealed, 1 L aluminum-foil bag, sized appropriately for GC analysis. Ten airbags (for simplicity, only four bags are illustrated) each feature a self-sealing structured polycarbonate (PC) stopcock straight valve and are connected to ten micro vacuum pumps via well-sealed tubes. Each pump has an inlet and an outlet. A Hydrophobic (PTFE) filter with a 0.45 $\mu$m pore size is attached to the inlet for prohibiting the dust. The outlet is tightly connected to the valve of the sampling bag, allowing collecting air when the valve is opened. All airbags are stored in a storage box to ensure the safety in case of strong wind. A GPS-receiver and a meteorological sensor (iMET XQ2) are used. iMET XQ2 is the second-generation sensor manufactured by International Met Systems, it is designed for UAV deployment, with a 5-hour rechargeable lithium battery and 15-hour data storage. It operates within a relative humidity range of 0-100 %, a temperature range of -90°C to +50°C, and a pressure range of 10 to 1200 hPa. It also provides GPS information, including time, longitude, latitude, and altitude. The whole procedure is programmable through a Micro Control Unit (MCU), and the sampling altitudes are pre-set before each flight.

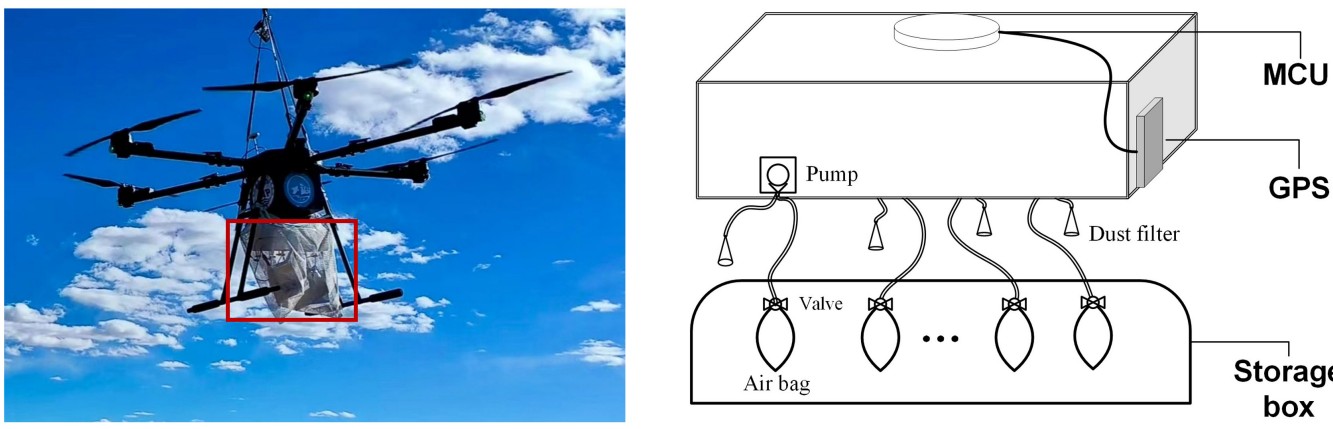

**Figure 1.** The design of the sampling system and its equipment on a UAV

## 2.2 Sampling procedure

The size of the gas collection system is $39\text{cm} \times 18\text{cm} \times 12\text{cm}$, and the total weight is 2.4 kg. The peak power of the sampling
is about 10.8 W. An extra 12 V small Lithium battery (capacity of 2 Ah, and about 150 g weight) is used to power the pump. Therefore, it can be carried by UAVs with sufficient capacity. small UAVs. The following operations are performed before each flight: bags must be flushed with high-purity nitrogen at least 5 times before sampling; each bag must be carefully labelled to register its logging information, such as time, location and altitude for future analysis; concern must be taken when mounting on UAV to prevent pollution from human activities. The working flow chart illustrated in Figure 2 provides a detailed view of
the process, including pre-processing, parameter configuration, and operational procedures.

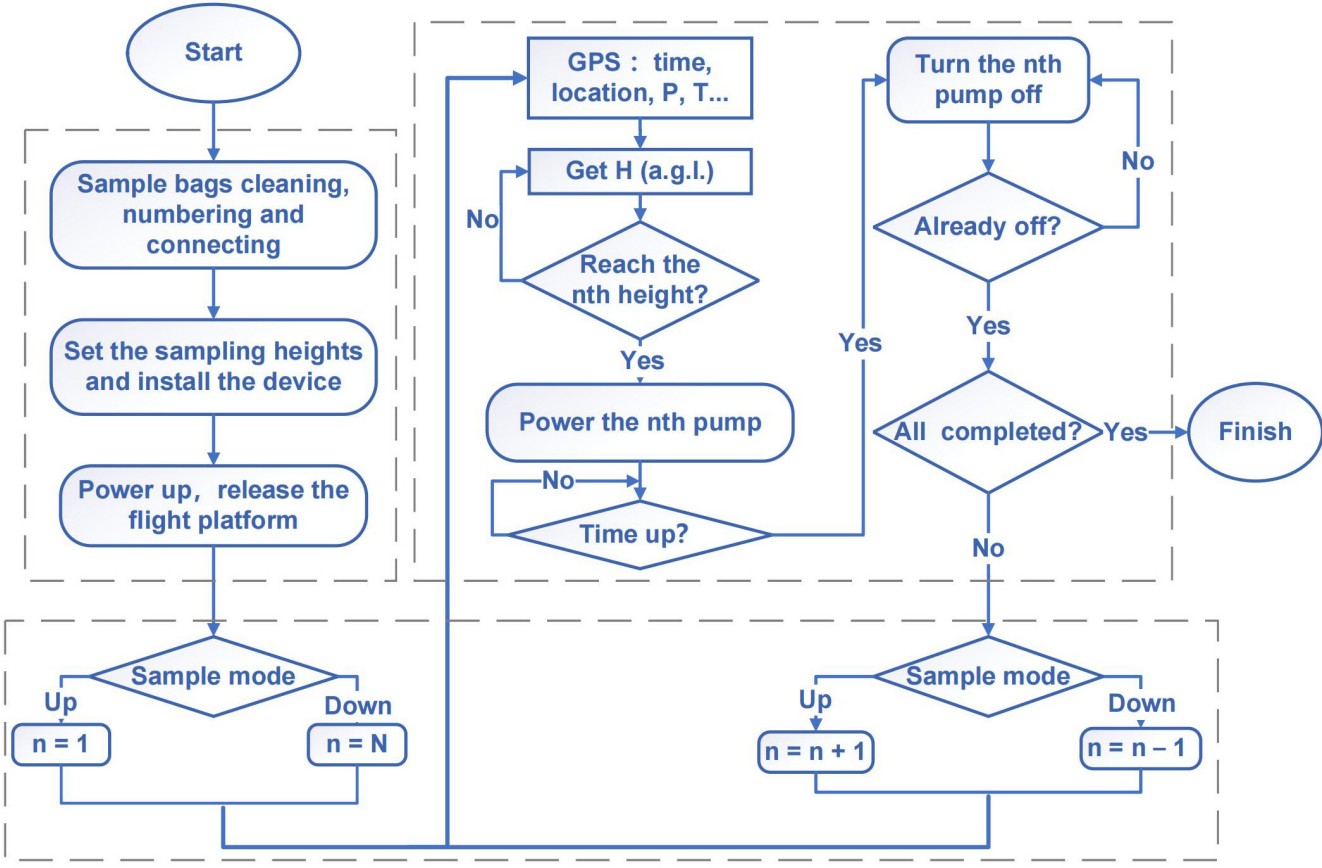

**Figure 2.** The working flowchart of the gas collection system

During the flights, the real-time altitude is calculated on-line at 1 Hz through pressure p and air temperature T collected from iMET XQ2 by:

$$Z = -\int_{p_0}^{p} \frac{RT}{g} \, d\ln p \tag{1}$$

where $p_0$ is the surface pressure, R is the ideal gas constant $287.05 \mathrm{J} \cdot (\mathrm{kg} \cdot \mathrm{K})^{-1}$, g is the gravitational acceleration as a constant $9.80665 \mathrm{m} \cdot \mathrm{s}^{-2}$. While it is also feasible to use the UAV's barometric and GPS-filtered altitude, we used the iMET XQ2 sensors to ensure consistency with the other atmospheric parameters being recorded and processed simultaneously.

Due to the mobility and flexibility of the UAV platform (Figure 1), it can be used as a self-operating instrument for vertical distribution observation of greenhouse gases. The sampling system supports two modes of operation—Up and Down. Up mode: the UAV ascends quickly at an average speed of approximately 4 m/s to a pre-set maximum altitude (e.g., 1300 m above ground

level). Due to energy efficiency considerations (Reuder et al., 2016), users need to hover at the target altitude to allow the pump to function (as shown in Figure 3a). The sampling motor runs for 11-20 seconds before stopping.Down mode: upon reaching the target altitude, the UAV hovers for about 10 seconds to collect the sample, after which the UAV automatically returns to its starting point with a slower descent (Figure 3b). We recommend Down mode as it requires less manual operation, conserves energy, and reduces extra sampling time. Our study collected only 15 samples (in the first two flights) during the ascent phase.

The operator manually controlled the loiter heights (as depicted in the height stage pattern in Figure 3a), presenting a challenge for the UAV to maintain stability. Each motor lasts 11-20 seconds and then stops. This sampling procedure repeats until the UAV lands on the ground, and the valves of airbags are closed. Above each valve, there is a sample cap with a silicone septum inside for syringe sampling.

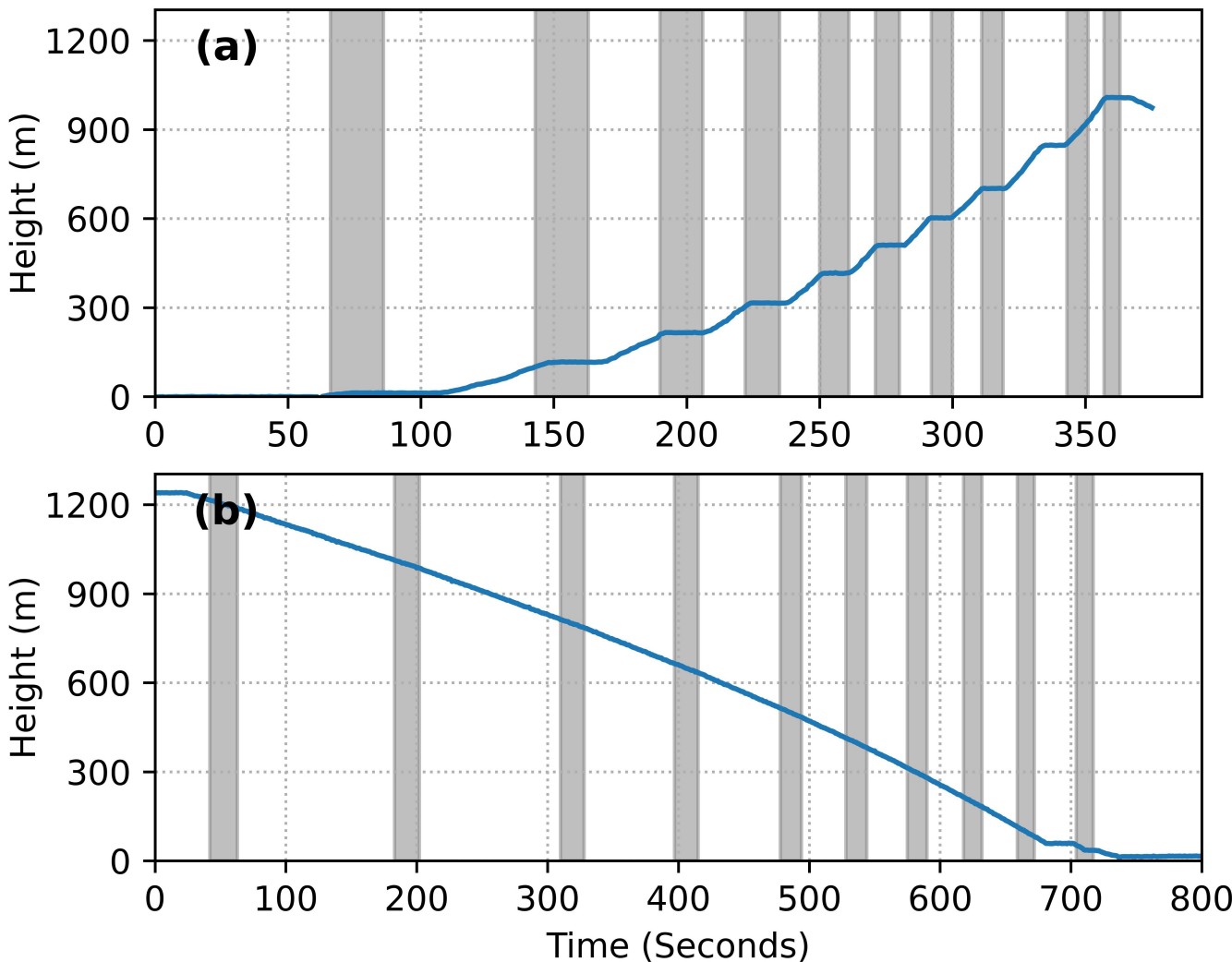

**Figure 3.** An example of sampling modes. The start times are (a) October 1 at 07:31, and (b) October 5 at 07:47(in local time). The lines indicate flight heights, while the gray shadows represent the operating times of each micro-motor.

## 2.3 Air sample analysis

The collected bags are measured with an Agilent GC 7890A (https://www.agilent.com.cn), and four GHG species ($CO_2$, $CH_4$, $N_2O$, $SF_6$) are simultaneously analyzed. The GC measurement is based on the fact that different components mixed in the samples flow at different speeds through the gas chromatography column, therefore, different gases are perfectly separated and accurately measured. We use a 13X molecular sieve (13XMS) to separate $CH_4$ and a Porapak Q for $CO_2$. Regarding $N_2O$ and $SF_6$, they are separated from $CO_2$ by the Porapak Q column and then backflushed to the detector. The GC is equipped with a

Flame Ionization Detector (FID) for detecting $CH_4$. $CO_2$ is converted to $CH_4$ using a nickel converter before being detected by the FID, as the FID only responds to carbon-containing organic compounds. Additionally, an Electron Capture Detector (ECD) is used for $N_2O$ and $SF_6$. For more information about the injector, gas line, valve-driving models, and laboratory accuracy testing, please refer to the details in our previous studies (Yuesi and Yinghong, 2003; Wang et al., 2010). The GC signals, mostly represented by area or peak due to gas absorption, are directly related and translated to gas concentrations, the signals are carefully calibrated with standard gases traced to the National Institute of Standards and Technology (NIST) scale. A linear regression is established between the area and the concentration of standard gases:

$$C = a \cdot \text{Area} + b \tag{2}$$

Where $C$ represents the concentration of the detected gas, *Area* represents the peak area of the detected gas, and $a$ and $b$ are coefficients given through calibration with standard gas. The standard gas is injected multiple times (n≥7), and the standard deviation of parallel determinations is calculated to determine the detection limit and precision using a specific formula. Each type of GHG is measured in terms of its volume mixing ratio (VMR). The precisions, represented by the coefficients of variation, are 0.18 % for $CO_2$, 0.99 % for $CH_4$, 0.22 % for $N_2O$, and 1.7 % for $SF_6$ at the average levels of 0.75 ppm for $CO_2$, 0.02 ppm for $CH_4$, 0.74 ppb for $N_2O$, and 0.20 ppt for $SF_6$. The detection limits of this method are 2.4 ppm for $CO_2$, 0.07 ppm for $CH_4$, 2.6 ppb for $N_2O$ and 1.5 ppt for $SF_6$.

## 3 Field Experiments

### 3.1 Sites

Field experiments were conducted at two high-altitude stations located in the Tibet Plateau:

(1) Cho Oyu basecamp (28.24°N, 86.59°E): This is a newly established temporary station without greenhouses measurements records yet. Its basecamp, located at 4,950 m a.s.l., serves as the starting point for the scientific research team to the summit of Mount Cho Oyu, which is about 8201 m a.s.l., the 6th highest mountain in the world.

(2) Qomolangma Station, CAS (28.36°N, 86.94°E): It is located at 4300 m a.s.l. and is on the northern slope of Mount Qomolangma (8848.86 m a.s.l., the highest mountain in the world). This station was established in 2005 by the Institute of Tibetan Plateau Research, Chinese Academy of Sciences (Ma et al., 2023).

Both sites are located in Tingri County, in Rikaze City, with detailed geographic location and elevation information provided in Figure 4.

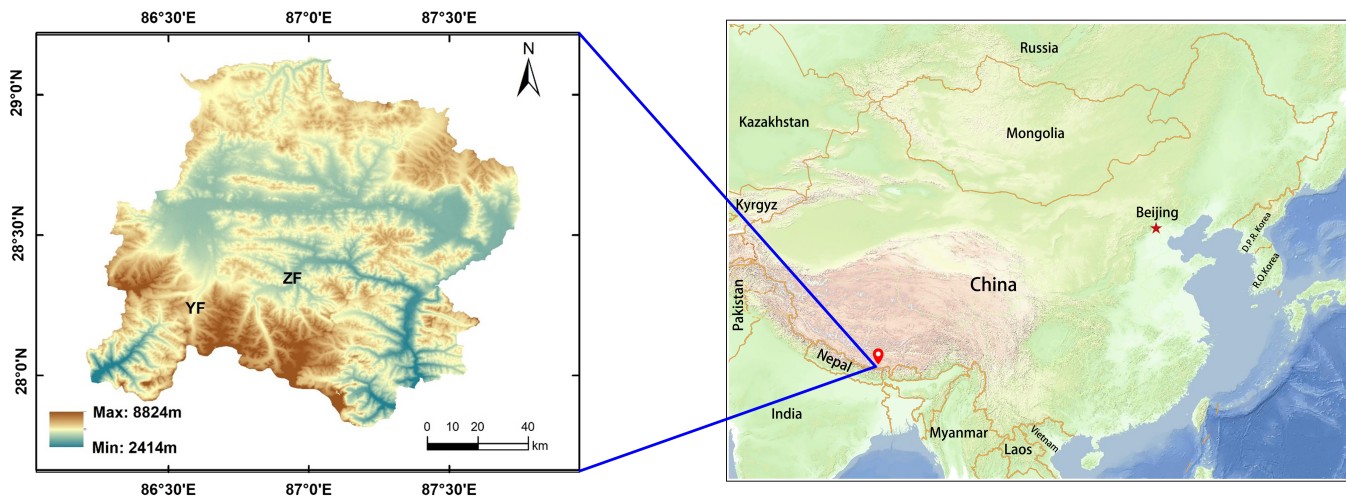

**Figure 4.** The experiment sites: YF corresponds to Cho Oyu base camp, and ZF corresponds to the Mount Qomolangma Station. The Digital Elevation Model (DEM) data is sourced from the Geospatial Data Cloud (http://www.gscloud.cn)

## 3.2 Results and analysis

Between 29 September and 03 October 2023, three flights were attempted in Cho Oyu, but only one flight succeeded due to bad weather conditions and MCU failures. On Oct.03, the system was transported to the Qomolangma Station, and 12 flights were successfully operated in the next 3 days.

During each flight, 10 bags were collected at 10 different altitudes, and it took about 40 minutes per flight. The flight and sampling information is listed in Table 1. In total, 139 samples were collected during the whole field campaign. The mean and standard deviation of the four greenhouse gases, as averaged across all samples, are listed in Table 2, showing low concentrations and minor variances.

**Table 1.** Sampling log of GHGs measurements during UAV flights in the Mount Qomolangma Region

| Site | Local Date | Local Time | Max height(m) | Number of Samples |
|------|-----------|-----------|---------------|-------------------|
|      | 2023/10/01 | 08:32 | 588.0 | 5 |
| YF   | 2023/10/02 | 07:31 | 1007.9 | 10 |
|      | 2023/10/03 | 11:53 | 1112.3 | 7 |
|      | 2023/10/03 | 15:35 | 1113.2 | 9 |
|      |            | 07:41 | 1113.8 | 10 |
|      |            | 09:38 | 1214.9 | 10 |
|      | 2023/10/04 | 11:28 | 1213.2 | 10 |
|      |            | 13:31 | 1212.9 | 10 |
| ZF   |            | 20:05 | 1214.4 | 9 |
|      |            | 07:42 | 1215.0 | 10 |
|      |            | 09:47 | 1213.5 | 10 |
|      | 2023/10/05 | 11:37 | 1203.5 | 9 |
|      |            | 13:43 | 1213.8 | 10 |
|      |            | 16:34 | 1211.7 | 10 |
|      |            | 20:36 | 1214.6 | 10 |

**Table 2.** Means and standard deviations of gas mixing ratios of all samples

| Site | Time | $CO_2$ (ppm) | $CH_4$ (ppm) | $N_2O$ (ppb) | $SF_6$ (ppt) |
|------|------|-----------|-----------|-----------|-----------|
| YF | 2023/10/01-03 | 421.13±4.76 | 1.98±0.01 | 337.38±1.26 | 11.86±0.56 |
| ZF | 2023/10/03-05 | 418.35±2.54 | 2.00±0.02 | 337.15±1.41 | 11.76±0.54 |

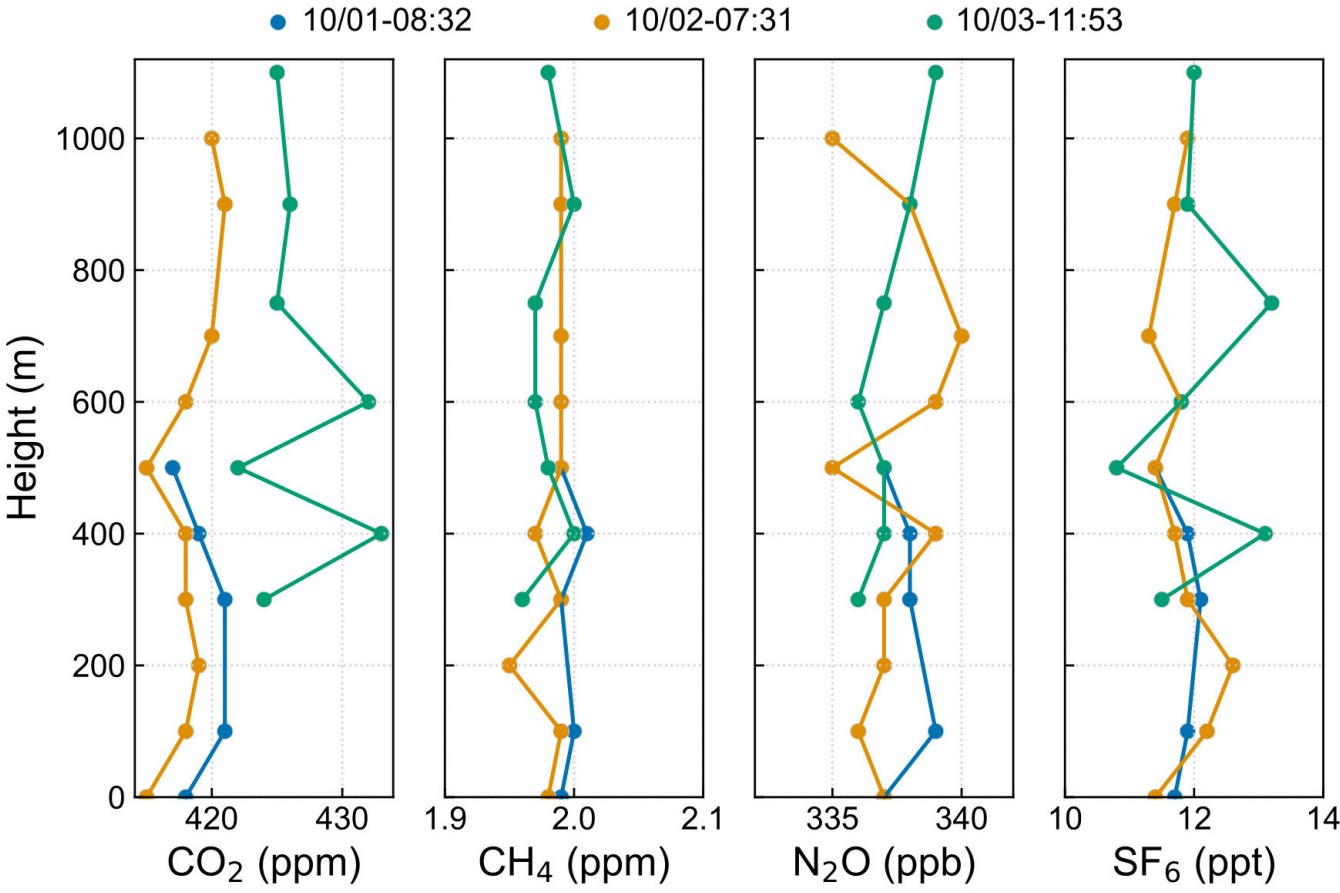

**Figure 5.** Profiles of 4 components ($CO_2$, $CH_4$, $N_2O$, $SF_6$) analyzed from Agilent GC 7890A and heights are measured by iMET XQ2 obtained in YF from 01 October to 03 October. The profiles in 01 October and 02 October are measured from ascent, and the profile in 03 October is from a descent.

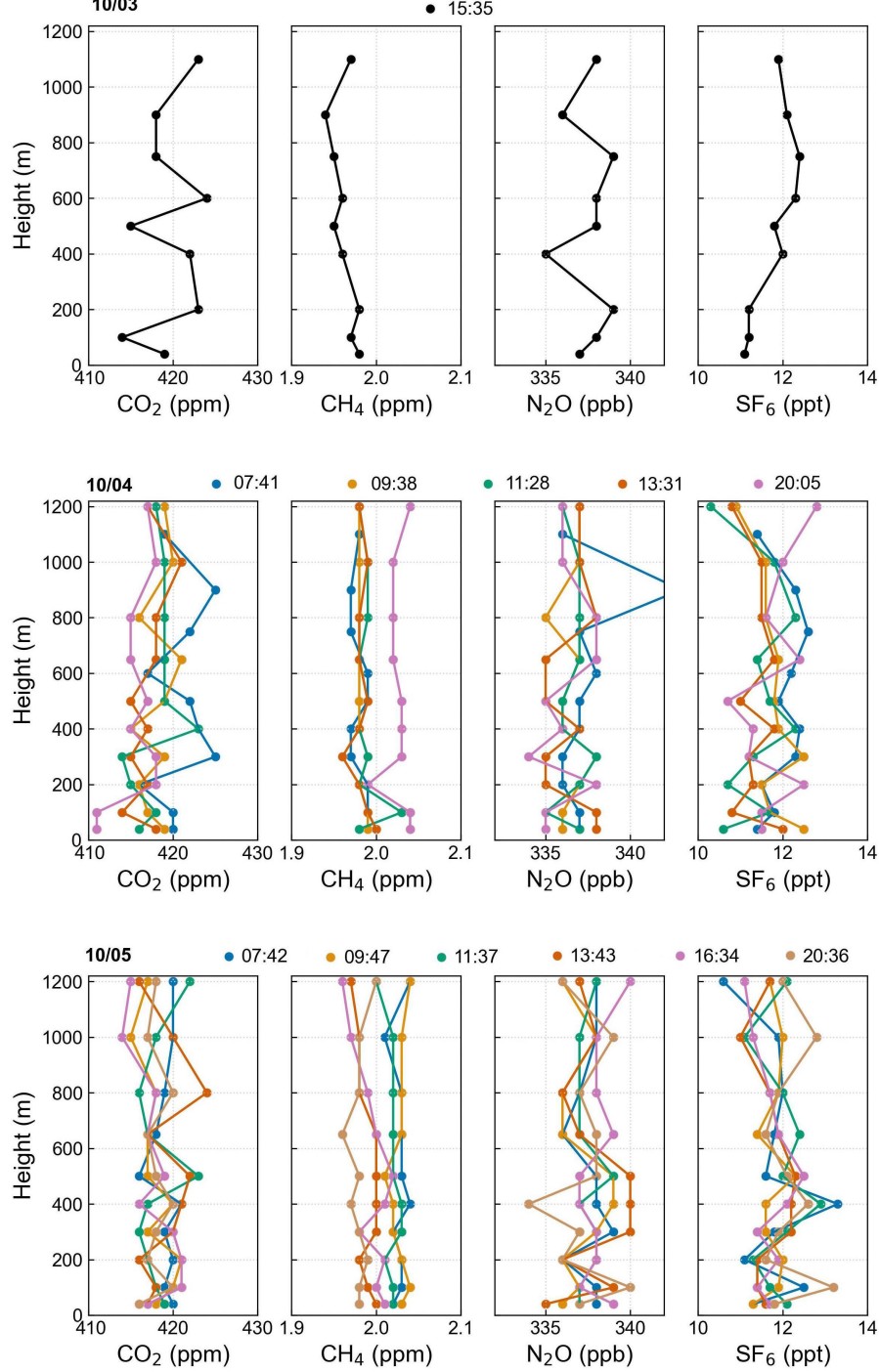

**Figure 6.** Same as Figure 5, but for ZF from 03 October to 05 October. The profiles are from descent.

The vertical distribution of four species is displayed in Figures 5 and 6 for the Cho Oyu site and Qomolangma Station, respectively, showing irregular fluctuations.

To examine the diurnal variation of GHGs, we compute the integral average of their mixing ratios. We also use ERA5 reanalysis data to determine the boundary layer height (BLH). This method allows us to categorize our samples into two distinct groups: those above the boundary layer and those below it. Figure 7 illustrates two time series of mixing ratios for four different species. The series for $CO_2$, $N_2O$, and $SF_6$ show a consistent pattern; however, the variations in $CO_2$ within the boundary layer height (BLH) are more pronounced than those above it. The downward trend observed on 04 October may reflect the intensification of natural processes due to sunlight and the increase in boundary layer height caused by solar heating. In contrast, $CH_4$ is well mixed; trends were inversely correlated with BLH and showed a slight increase on the night of 10/04 compared to the daytime. The increase in $CH_4$ levels exceeded the relative standard deviation (RSD) of our equipment, which may be attributed to local livestock or meadow emissions. Due to limited observational data and insufficient information on emission sources and meteorological conditions, accurately quantifying and assessing the contributions of these factors is challenging.

## 4    Conclusions

In this study, we developed a simple vertical stratified atmospheric sampling device that can be mounted on a middle-size UAV, a tethered balloon, or the roof of an electrical car, which can get air samples at different altitudes during a single flight, either liftoff or return. After the collection is completed, the gas bag is closed to facilitate subsequent chromatography analysis to obtain the concentration of atmospheric components at multiple atmospheric altitudes. At the same time, the device can record the temperature, pressure, humidity, and location of each layer of the atmosphere.

In conclusion, the device has the following advantages: 1) its flexible design and adaptability make it suitable for integration with a variety of analytical instruments, enabling three-dimensional monitoring across diverse platforms; 2) its cost is less than US $ 5000, supporting widespread deployment and facilitating broader adoption in diverse research settings; 3) once the MCU is pre-set before the flight, its automatic operation and quick response time ensure simplicity and ease of use. As a result, this device is suited for extended periods of atmospheric observation and is minimally affected by terrain.

A 5-day continuous observation campaign was conducted at the Cho Oyu Base Camp and Qomolangma Station. We integrated the sampling system into a medium-sized hexacopter UAV platform and obtained 15 GHG vertical profiles up to 1215 m. The temporal scope of the measurements, although informative, constrains the degree to which long-range transport may have affected the observed trends. Greenhouse gases like $CO_2$ exhibit more pronounced variations within the boundary layer, while $CH_4$ levels rise slightly at night, potentially due to local emissions. This nocturnal increase in $CH_4$ could be linked to reduced atmospheric mixing during lower BLH, which leads to the accumulation of emissions near the surface. To enable continuous atmospheric monitoring (Kunz et al., 2018; Reuter et al., 2021), we still need to reduce equipment weight for easier long-term deployment. It also helps us assess the distribution of greenhouse gases, elucidate their sources and sinks, and disentangle the signals from local vertical mixing to long-range transport. It also has the potential to provide the prior value of vertical distributions of GHGs to calibrate and evaluate the satellite retrievals over complex topography. Comprehensive satellite observations (such as OCO-2, GOSAT, and Sentinel-5P) within a 100 km radius did not cover our region of the campaign. However, we have limited $XCO_2$ data from OCO-3 that aligns with our measurements. On 01 October 06:11 UTC, the $XCO_2$ values were 416.00 ppm at Qomolangma and 412.64 ppm at Cho Oyu. On 05 October 04:36 UTC, they were 417.96 ppm at Qomolangma and 414.66 ppm at Cho Oyu.

Although using UAVs or balloons to monitor or inspect various sites has proven to be useful, this method has its limitations, including a relatively low sample resolution, as only 10 samples are collected. This results in a much coarser atmospheric profile, which is more challenging to relate to the atmospheric boundary layer (ABL) cycle. Additionally, the weight of the sampling device poses a challenge for smaller UAVs, making it less feasible for lightweight platforms. Adverse weather conditions such as strong winds, can interfere with the safe flight of these devices and the GPS signal. Furthermore, it is not advisable to use this technique for monitoring chemically active gas components in case the gas content changes during transportation. To address these issues, we will continue to optimize the design of the device to improve its performance and adaptability. We expect it to be used in a wider range of applications, such as understanding the sources and formation mechanisms of air pollution events.

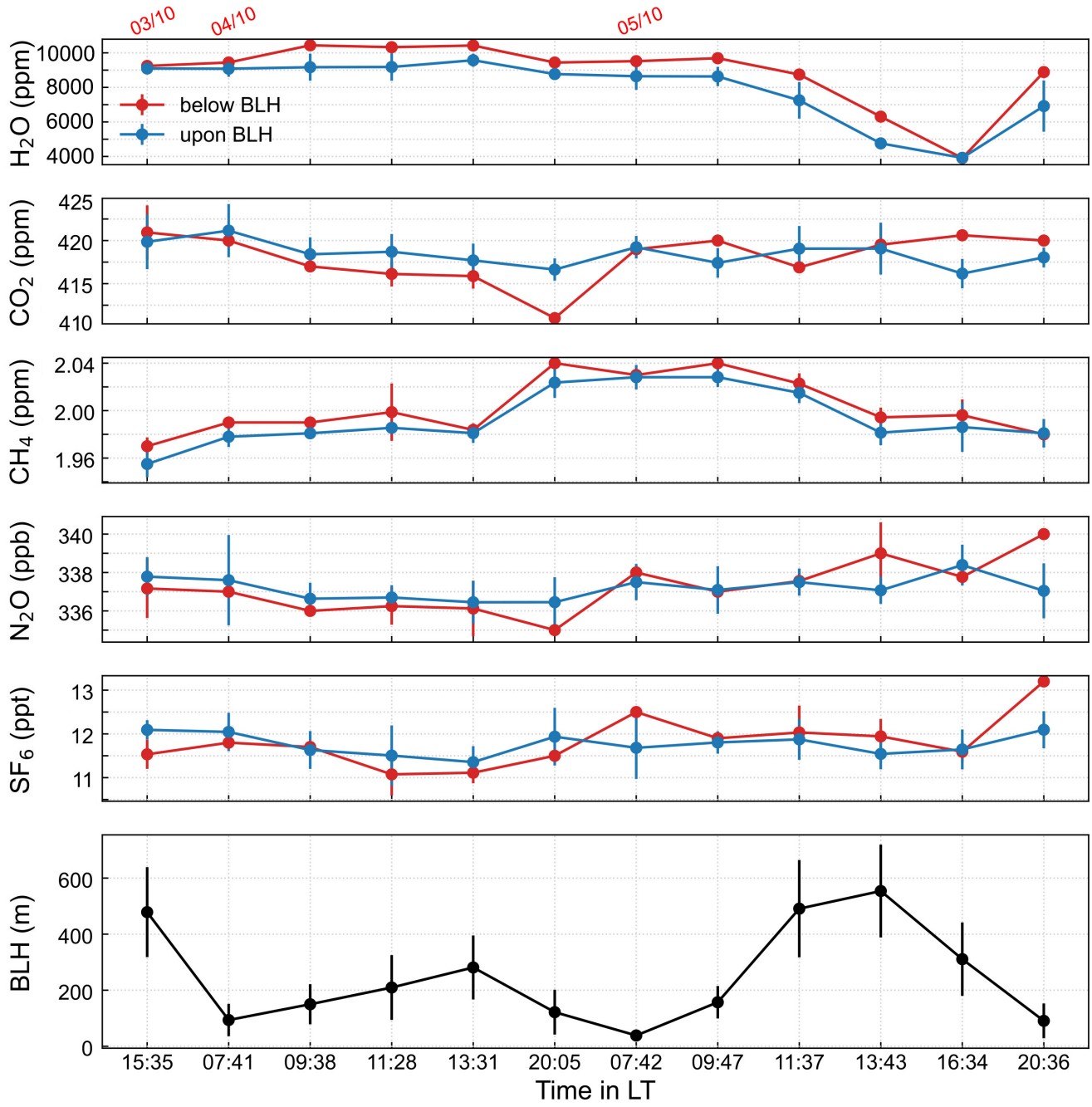

**Figure 7.** Time series of 5 components ($H_2O$, $CO_2$, $CH_4$, $N_2O$, $SF_6$) taken in ZF from 03 October to 05 October and collocated boundary layer height data from ERA5, utilizing the integrated average value for each observation. Error bars represent the dispersion (standard deviation) of values across space.

*Data availability.* The observation data are available upon request from the corresponding author(dmz@mail.iap.ac.cn). ERA5 data used in this study are accessible from the ECMWF web page: https://www.ecmwf.int (last access: 6 October 2024; Hersbach et al. (2020)). OCO3 data are downloadable from https://ocov3.jpl.nasa.gov/ (last access: 1 November 2024; Taylor et al. (2020)).

*Author contributions.* YZ, CQ, and MD contributed to the manufacturing of the sampling system. YW, MD, MZ, CQ, and YZ designed and conducted the campaign. YW, YZ, and CQ carried out the laboratory analyses, and YZ, MZ, MD, YW, and XT contributed to the preparation of the manuscript.

*Competing interests.* The authors declare no conflicts of interest.

*Acknowledgements.* This work was supported by the National Natural Science Foundation of China (42030107), and the Second Tibetan Plateau Scientific Expedition and Research Program (2022QZKK0101)

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
