# Peer review of "Observation of GHG vertical profile in the boundary layer of the Mount Qomolangma region using a multirotor UAV"

_EGUsphere, 2024_

## Author Response (AR1)

*Thank you very much for taking the time to review this manuscript, your constructive contribution helped to improve the quality of the paper in both sciences and writing, making it more valuable, and easily understood for the readers. Please find the detailed responses below and the corresponding revisions in track changes in the revised version of this manuscript.*

**Anonymous Referee #1**

The **black italic** texts are comments of the reviewer, and the **red italic** texts are responses.

**Comment***s: Dear authors, congratulations on an exciting work. I believe that with minor adjustments, your article can be improved to enhance its impact. The main adjustments I believe are necessary are a contextualization of this research in the larger body of UAS-based GHG measurement literature, more details on the calibration and use of the iMET XQ2, and more details on the impact of the aircraft on sampling method (up vs. down). Below are a few recommendations (combining broad and detailed):*

**Introduction/Motivation/Conclusion:** *In your argument, you juxtapose in-situ and remote measurements, and within in-situ technologies, you highlight the limitations of towers and manned aircraft to set your reason for developing this technology. However, you fail to contrast your technology with previous UAV-based GHG sampling. I believe the readers would greatly benefit if you were able to highlight the differences between your method and prior methods, for example:*

*- https://amt.copernicus.org/articles/11/1833/2018/*
*- https://amt.copernicus.org/articles/15/5599/2022/*
*- https://amt.copernicus.org/articles/14/153/2021/*
*- https://amt.copernicus.org/articles/17/677/2024/*

*I say this because each method has various advantages and disadvantages, and readers interested in your method should clearly know the cases in which your method works best, as well as its limitations. For example, Kunz and Azevedo (linked above) have much lighter, cheaper, and higher spatial resolution technologies than yours. However, their technologies are limited to one or two gases, whereas yours is a multi-gas solution (which is more straightforward to compare with satellites and towers). Nonetheless, your 10 samples produce a much coarser atmospheric profile that is harder to relate to the ABL cycle.*

*My comment here is further corroborated by your conclusion, where you emphasize the low cost and lightweight aspects of your solution, which is relatively not entirely correct (see more comments below).*

*So, please add a paragraph in your introduction or method contextualizing your work within the UAV-based GHG sampling literature, highlighting the advantages and best applications of your method, as well as its limitations.*

**Response:** *Thanks for the comments and the references provided above, we have added a paragraph in section introduction and more works related to UAV-based measurements are referred and listed. In the added paragraph, we explicitly compare our method with existing UAV-based GHG measurement methods, highlighting its advantages and noting its limitations. The new added paragraph is shown below, please also see the track marks in the revised manuscript.*

*'Many works have used UAVs for in-situ observation of GHGs, primarily utilizing Non-Dispersive Infrared (NDIR) sensors to measure CO2 and CH4(Kunz et al., 2018; Reuter et al., 2021; Britto Hupsel De Azevedo et al., 2022; Han et al., 2024). While NDIR and other low-cost sensors have the advantage of real-time and continuous monitoring due to their lightweight design, they face challenges such as frequent calibration requirements and potential fluctuation due to the change of ambient environments such as pressure, temperature and also vapor content in the variable atmosphere along with altitude(Liu et al., 2022). In contrast, flask(usually made of metal) analysis allows air samples to be collected and analyzed in a controlled laboratory(Loftfield et al., 1997), but flask evacuation and cleaning is more labor-intensive, and it is not easy to operate in-flight measurements. We have developed a device similar to flask sampling but aluminum bags are used, featuring a relatively lighter design, and expanded its capabilities to analyze additional GHG components. Note that our system requires a higher payload capacity and larger platform size than online analysis sensors. '*

*Comments:*

**Line 69-70:** *You say your payload has 2.4 Kg, and it can easily be carried by any small UAV. Because the terms easily and small are relative, this can cause confusion. Within UAV work, 2.4 Kg is not customarily considered small/light. In fact, very few commercial multirotors have this payload capacity. Within the multirotors that have this capacity, most would only be able to carry it for less than 15-20 minutes. Therefore, I recommend you eliminate the adjectives "small" and "easily". Just say that the payload is vehicle-independent and can be used on any multirotor capable of handling this size and weight. If you think making the point that many other vehicles could carry it is necessary, give the reader at least five different examples of commercial vehicles with such capability (I believe you will struggle to find more than 3 that meet your requirements).*

**Response:** *Thanks for the comments. In the revised manuscript, we eliminated the adjectives "small" and "easily."*

*You are right, our sampling device is not as light as people expected, and cannot be carried by a commercial light-weight UAV. In operation, we used a middle-sized UAV, which can carry 20~30 kg payloads, the size of the UAV is about 900mm(W)\*900mm(L)\*800mm(H), and two 37Ah batteries are used in operation to ensure that it can go up 1300m above ground and fly more than 80 minutes, the 14s battery is 60V full of charge. we rewritten the sentences and a detailed description of the UAV is given in the revised manuscript.*

*- Additionally, in UAV work, payloads are often considered in terms of SWaP (size, weight, and power). So, I recommend you also add the power consumption of your system (in Watts) because that can also limit platform selection. For example, do the pumps operate at 5 or 12V? If it is 12V, you can't use a UAV that uses a 3S battery.*

**Response:** The sampling device is about 39cm x 18cm x 12cm, and the weight is about 2.4kg. The peak power of the sampling is about 10.8w. An extra 12V small Lithium battery (capacity of 2Ah, and about 150g weight) is used to power the pump. The battery can work for more than two hours and runs for the 10-sampling twice.

*- Another comment. Since you are driving the point that the measurement technology can be used in other vehicles, it implies that others can use it. So, it begs the question, is it open source? Is it available to the larger community? If it is, you should mention it because it makes the vehicle independence argument more important.*

**Response:** Sorry for the misunderstanding, we want to say that, other than UAVs, this sampling device can be used for a Tethered balloon, or on top of an electrical car. We have rewritten the sentences in the revised manuscript.

*- Finally, as you experienced, UAVs capable of carrying 2.4 Kg often can't handle wind speeds larger than 15 m/s, which is a considerable limitation for folks considering adopting this technology for year-round GHG monitoring (say, one flight per day, every day), or as you put it "a new test-bed for long-term and continuous (...) monitoring." Therefore, 2.4 kg is not small or easy.*

**Response:** Sorry for the misunderstanding, yes, you are right that UAV is not an all-weather platform, especially for small- and middle-size UAVs or Tethered balloons. For security, we only let it fly when the wind is under 15m/s. We have rewritten the sentences in the revised version to make it clear. Thank for the comments.

*- - This tone adjustment should also be reviewed in your conclusion.*

**Response:** Thank for the comments, all the above concerns have been mentioned in section "Conclusion".

*Figures (all): they are all too small and unreadable in print. I hope this is an artifact of this pre-print format. If not, be sure to increase them in the final paper.*

**Response:** Thank for the comments, during the conversion from latex to pdf, the size of some figures is reduced, which is improved in the revised manuscript.

*Comment:*

***Line 75:*** *What is the motivation to calculate altitude from the iMET XQ2 sensors? Was it not possible to use the UAV's barometric+GPS filtered altitude?*

***Response:*** *We suggest calculating altitude from the iMET XQ2 sensors because GPS sometimes lost signal, while pressures and temperature can be continuously recorded, particularly under challenging topographic conditions in Tibetan area. More sentences have added in the revised manuscript to explain the motivation to calculate altitude from the iMET XQ2 sensors.*

*iMET XQ2: I understand this paper focuses on GHG and not atmospheric boundary layer (ABL) measurements. However, since you chose to correlate your measurements with ABL behavior and chose to use the XQ2 as your source for altitude measurements, the following comments are critical for the scientific relevance of your article:*

*- The XQ2 is a notoriously bad UAV PTU solution because it does not account for solar shielding, UAV-based heat sources, sensor air flow minimums, and other measurement interference sources.*

***Response:*** *Thanks for the very important and constructive comments. We thought that all the temperature sensor are coded for shielding the solar radiation, and never recognized and surprised this is not the case for iMET XQ2. When you pointed out, we wrote a letter to the manufacture and got the answer just as you say.*

*While we only use iMET XQ2 data to calculate altitude as the GPS signals sometimes lost due to the complex Tibetan topographic conditions. The altitudes, calculated from XQ2 based on the equation 1 of the manuscript, are compared with that of GPS as shown in figure S1. The difference between the two data are less than 7 meters. As given in equation 1, the calculated altitude is dependent on the temperature difference between the ground and specific altitude. The small difference attribute to the relative short time flight (about 40 minutes from take-off and landing, and the 10-altitude sampling works during the landing period, which is less than 30 minutes), which lead to small temperature difference uncertainty and the results is reasonable and can be used for profile analysis of GHGs.*

*Thanks for the comments, as discussed above, the profile analysis is trustable, while big differences in in XQ2 temperature, therefore, we delete the content related to the ABL analysis and equation 3.*

[Figure]

Figure S1. Comparison of GPS height (a.g.l) and the height calculated from iMET XQ2, alongside the measurement of iMET XQ2 during flight starting on October 5th at 16:34 local time.

- *Given your extensive use of its information, it is necessary that you provide the reader with information on how you calibrated it and integrated it into the platform. Otherwise, it will be harder to trust your data.*

**Response:** *Thanks for the comments, as discussed above, the profile analysis is trustable, the content related to the ABL analysis are deleted.*

- *For example, the PT-100 bead thermistor on the XQ2 requires at least a 5 m/s flow over it. However, its 1-second time-response limits its flight speeds for good ABL reconstruction to 2.5 m/s (this issue is often solved with independent aspiration fans). The same is true for the HYT-271 hygrometer inside the XQ2. For good references on why placement, aspiration, and operation UAV for reasonable PTU measurements, you can take a look at other papers on this matter:*
- *https://amt.copernicus.org/articles/11/5519/2018/*
- *https://www.mdpi.com/1424-8220/19/6/1470*
- *https://journals.ametsoc.org/view/journals/atot/35/8/jtech-d-18-0019.1.xml*
- *https://www.mdpi.com/1424-8220/19/9/2179*

**Response:** *Thanks for the constructive comments, we have deleted the contents related to ABL analysis (Line 124 to Line 138, and Line 147-149) in the revised manuscript.*

**Comment:**
**Line 80:** *What is "Just Go"? I suggest explaining it in words or substituting it with an actual technical term.*

**Response:** *Thanks for the comments. We want to say this device is easy to set up and portable to be taken for field measurements. We have replaced this word and rewritten the sentences in the revised manuscript.*

**Comment:**
Line 82: *Here, you explain your procedure for sampling during the descent. However, figure 2 and the conclusion allude to data collection during the ascent. Given that you are producing a relatively "low" resolution profile (with samples at approximately 100 meters), this is not a problem for considerations regarding propeller layer mixing for the gas samples. Nonetheless, it will considerably impact your PTU measurements and potentially your Z calculation. Therefore, you should make these limitations more explicit and describe your system's best use (up or down).*
*- For a resource on the limitations of ABL PTU measurements on ascent versus descent, I recommend this paper https://amt.copernicus.org/articles/9/2675/2016/.*
*- This comment should also affect the tone of your conclusion.*

**Response:** *Thanks for the comments and sorry for the misunderstanding. Yes, the system can be set to work during ascending or descending time, we tried to say this device is easy to set up and portable to be taken for field measurements. As discussed, we only use iMET XQ2 data to calculate altitude as the GPS signals are sometimes lost due to the complex Tibetan topographic conditions and the short working time, which is less than 30 minutes, the altitude differences due to temperature uncertainty are less than 7 meters and it is trustable and can be used for profile analysis of GHGs. More words have been added to make it clear in the revised manuscript.*

[Figure]

*Figure 3 An example of sampling modes. The start times are (a) October 1 at 07:31, and (b) October 5 at 07:47(in local time). The lines indicate flight heights, while the gray shadows represent the operating times of each micro-motor.*

**Comment:**
**Line 87:** *"Angilent" is misspelled.*
**Response:** *Thanks for the comments and sorry for the typo, we have corrected it in the revised manuscript.*

**Comment:**
*Line 116: I believe adding one more word, such as "at" or "by," to the sentence "Institute of Tibetan Plateau Research, the Chinese Academy of Sciences" might make it better for readers not familiar with the relationship between these two institutions.*

**Response:** *Thanks for the comments, "the" word should be deleted, we corrected it as a traditional and official permit in the revised manuscript.*

**Comment:**
**Line 117:** *I believe adding one more word, such as "near" or "at," to the sentence "located in Tingri County, Rikaze City" might make it better for readers unfamiliar with the local geography.*

**Response:** *Thanks for the comments, we have replaced it with "located in Tingri County, in Rikaze City", and we added a map to demonstrate its location in Figure 4.*

*Comment:*

*Table 2: Please provide more information about how this mean is calculated in the text (a simple one-liner). It is unclear if it is the mean for the whole dataset (all samples, flights, and days). If it is, what is the relevance of this mean? It seems it only indicates that GHG variations at the sites are minor (compared with sites at lower altitudes or near urban centers). For example, near urban centers, CO2 measurements can show up to a 30 ppm gradient (surface to top of profile at 1500 m) in a flight due to a low altitude inversion in the ABL.*

**Response:** *Thanks for the comments, more words were added to clarify in the revised manuscript.*

*"The mean and standard deviation of the four greenhouse gases, as averaged across all samples, are listed in Table 2, showing low concentrations and minor variances."*

*Comment:*

**Line 129:** *I believe there is an editing mistake here. How are potential temperature and specific humidity derived from GPS? Were you not using PTU to calculate Z?*

**Response:** *Thanks for the comments, we have deleted these sentences in the revised manuscript, as there may exist big uncertainties due to the inaccuracy of XQ2 temperature, Figures 5 and 6 are also deleted. By the way, Z is calculated from XQ2, not GPS, sorry for the mistake.*

*Comment:*

*Figures 4 - 7: In the methodology, you mentioned the data was gathered during the descent, but later you mention data from the ascent. Were all the data points in Figures 4 -7 only collected during the descent?*

**Response:** *Thanks for the comments, and sorry for the unclear statement. The sampling for 01 October and 02 October 2023 were made during ascent, while on 03 October 2023, they were collected during descent time. more words are added in the revised version to make it clear.*

*Comment:*

**Line 157:**

*- As detailed above, this is not a lightweight system.*

*- Either I missed it, or you never mentioned the cost of your system. So why are you concluding it is a low-cost system? Given its dependence on a US $ 11k machine (GC 7890), it is costly compared to the other systems detailed above (which cost less than US $ 300).*

*- Granted, these solutions have fewer gases and lesser accuracies, although they have much higher spatial resolution.*

*- Therefore, I would rephrase this conclusion to say it is platform-independent (making it flexible) and less expensive than other solutions for the same gases with the same measurement accuracy (if that is actually true. I am unsure).*

**Response:** *Thanks for the comments, we agree with the reviewer's suggestion to revise the term "lightweight" to better reflect the system's capabilities, which allows it to be adapted for use with various UAV platforms capable of handling the specified payload.*

*For the sampling device only, the cost is less than US $5000. The sampling bags collected are sent to be analyzed in center where has GC already. We did not count in the expense of getting a new GC, all the work is supposed we have one yet, or it can be if not, the bag can be sent to any center where can do this, the cost for analysis of each sample bag is about $5, it is cheap too.*

*The analysis is not limited to GC-based analysis, it can also be measured with a calibrated Picarro or LGR analyzers, depending on the requirements of the study. We chose GC for this study due to we have one in our laboratory and multiple GHG species ($CO_2$, $CH_4$, $N_2O$ and $SF_6$) can be measured simultaneously by it.*

**Comment:**
*All in all, this is fascinating work. Congratulations. I hope my comments have provided you with resources to make it even better for the community in the final publication. Best of luck.*

**Response:** *Thanks for all your detailed comments and resources provided, which have expanded our knowledge of UAV measurement. Your constructive contribution helped to improve the quality of the paper in both sciences and writing, making it more valuable, and easily understood for the readers. Thank you very much.*

*The following papers are referred to and added to the list of section References.*

REFERENCES

Britto Hupsel de Azevedo, G., Doyle, B., Fiebrich, C. A., and Schvartzman, D.: Low-complexity methods to mitigate the impact of environmental variables on low-cost UAS-based atmospheric carbon dioxide measurements, Atmos. Meas. Tech., 15, 5599-5618, 10.5194/amt-15-5599-2022, 2022.
Han, T., Xie, C., Liu, Y., Yang, Y., Zhang, Y., Huang, Y., Gao, X., Zhang, X., Bao, F., and Li, S. M.: Development of a continuous UAV-mounted air sampler and application to the quantification of CO2 and CH4 emissions from a major coking plant, Atmos. Meas. Tech., 17, 677-691, 10.5194/amt-17-677-2024, 2024.
Hubbard, K., Lin, X., Baker, C., and Sun, B.: Air temperature comparison between the MMTS and the USCRN temperature systems, Journal of Atmospheric and Oceanic Technology, 21, 1590-1597, 2004.
Kunz, M., Lavric, J. V., Gerbig, C., Tans, P., Neff, D., Hummelgård, C., Martin, H., Rödjegård, H., Wrenger, B., and Heimann, M.: COCAP: a carbon dioxide analyser for small unmanned aircraft systems, Atmos. Meas. Tech., 11, 1833-1849, 10.5194/amt-11-1833-2018, 2018.
Liu, Y., Paris, J. D., Vrekoussis, M., Antoniou, P., Constantinides, C., Desservettaz, M., Keleshis, C., Laurent, O., Leonidou, A., Philippon, C., Vouterakos, P., Quéhé, P. Y., Bousquet, P., and Sciare, J.: Improvements of a low-cost CO2 commercial nondispersive near-infrared (NDIR) sensor for unmanned aerial vehicle (UAV) atmospheric mapping applications, Atmos. Meas. Tech., 15, 4431-4442, 10.5194/amt-15-4431-2022, 2022.
Loftfield, N., Flessa, H., Augustin, J., and Beese, F.: Automated gas chromatographic system for rapid analysis of the atmospheric trace gases methane, carbon dioxide, and nitrous oxide, Journal of Environmental Quality, 26, 560-564, 1997.

Reuder, J., Båserud, L., Jonassen, M. O., Kral, S. T., and Müller, M.: Exploring the potential of the RPA system SUMO for multipurpose boundary-layer missions during the BLLAST campaign, Atmos. Meas. Tech., 9, 2675-2688, 10.5194/amt-9-2675-2016, 2016.

Reuter, M., Bovensmann, H., Buchwitz, M., Borchardt, J., Krautwurst, S., Gerilowski, K., Lindauer, M., Kubistin, D., and Burrows, J. P.: Development of a small unmanned aircraft system to derive CO2 emissions of anthropogenic point sources, Atmos. Meas. Tech., 14, 153-172, 10.5194/amt-14-153-2021, 2021.

**Anonymous Referee #2**

*Greenhouse gases such as CO2 and CH4 are thought to be the primary human activities contribute to the current global warming, therefore many efforts, such as ground-based and space-based measurements along as the flux modelling, have been done to figure out the amount of such contributes. Those measurements focused on the column integrated amount, not the vertical profiles. Based on sampling method and UAV, the authors provided a simple and economic method for vertical profile of four GHG species (CO2, CH4, N2O, and SF6) in remote and inaccessible Tibetan area. CO distribution of but less measurements. The work is exciting and encouraging for the research of "Carbon" source and sink, and introduces an automatic low-cost and user-friendly multi-altitude atmospheric sampling device that can be mounted on small and medium-sized unmanned aerial vehicles, balloons, and other flight platforms to collect air samples at multiple altitudes. A five-day continuous observation campaign was conducted at Mount Cho Oyu Basecamp and Mount Qomolangma Station to analyze and explore the vertical distribution characteristics of four greenhouse gases. These measurements are critical for elucidating their sources and sinks, transport pathways, and influence on Earth's radiative balance, as well as for enhancing predictive capabilities for climate change. Overall, the article is well-structured, provides valuable insights, and language well-written. Further clarification can be made in some areas before published, and specific comments are as follows.*

**Comments:** *The innovative aspects of the study can be more explicitly emphasized. Additionally, the structure of the article should be introduced at the end of the introduction.*

**Response:** *We agree with the reviewer that the innovative aspects of our study could be more explicitly emphasized. In the revised manuscript, we have expanded the introduction to highlight the novelty of using a multirotor UAV for vertical profiling of GHGs in such a remote and challenging environment. We have emphasized the development of the sampling system and its potential applicability to other remote regions.*

*Following the reviewer's suggestion, we have added a brief overview of the manuscript structure at the end of the introduction. This provides readers with a clear roadmap for the study.*

**Comments:** *What impact does the change in BLH have on the vertical distribution and concentration of greenhouse gases?*

**Response:** *Thanks for the comment. We had 12 flights during three-day experiments from 01 to 03 Oct., 2023, the observed variations suggest that the weaker uplifting of BLH has very inconspicuous effect on the mixing ratio of GHGs. On 04 Oct., the BLH is notably lower than that of the adjacent days(Figure S2), and higher CH4 mixing ratio was observed(see Figure S3) in the afternoon of that day, this may attribute to local livestock or meadow emissions and the lower BLH.*

[Figure]

Figure S2. Variation of BLH with time

[Figure]

Figure S3. A heatmap BLH and mixing ratios of greenhouse gases.

*Comments: Does the vertical distribution of greenhouse gas concentrations change due to potential long-range transport?*

*Response: Thank you for your thoughtful question regarding the potential impact of long-range transport on the vertical distribution of greenhouse gas (GHG) concentrations.*

*Due to short term (five days) flight, we focused on capturing the local vertical profiles of GHGs at the Mount Qomolangma and Cho Oyu basecamp. The measurements were designed to reflect the local atmospheric conditions, with an emphasis on understanding the distribution of GHGs in the boundary layer over this brief period. This limited temporal scope reduces the likelihood of significant influence from long-range transport, as GHG concentrations at these sites are more likely to be shaped by local sources and meteorological conditions than by distant emissions.*

*We added a sentence in Line 163 to clarify:*

*'The temporal scope of the measurements, although informative, constrains the degree to which long-range transport may have affected the observed trends.'*

*Comments: "Figure 5. Same as Figure 5⋯" confusing.*
*Response: Thanks for the comment, this kind of typo errors from has been corrected in the revised manuscript.*

*Comments: "The conclusions of the article need further in-depth discussion.*
*Response: We added more sentences for the impact of BLH on GHG profiles, and the applicability of this sampling device are also discussed.*

*Comments: The text in figures is relatively small and needs to be improved.*
*Response: All figures have been reformatted to ensure readability in print and digital formats. Especially in the text-heavy Figures 1 and 2, we have simplified the text and increased the font size.*

*Comments: Line 13-20: Reference support required.*
*Response: Thanks for the comment. References and descriptions related to these references have been added to support the statements:*

*'Contemporary global warming...since the Industrial Revolution.'(Masson-Delmotte et al., 2019; Friedlingstein et al., 2023).*
*'Variations in emissions from natural and anthropogenic sources and atmospheric circulation patterns result in significant differences in greenhouse gas concentrations at different altitudes. '(Ren et al., 2011; Xie et al., 2013; Carnell and Senior, 1998)*

*Comments: Line 65: give the full name of iMET XQ2 and its main parameters*
*Response: Thanks for the comment. We added sentences related iMET XQ2:*
*'iMET XQ2 is the second-generation sensor manufactured by the International Met Systems, it is designed for UAV deployment, with a 5-hour rechargeable lithium battery and 15 hours*

*long of data storage. It works for relative humidity of 0-100%, for temperature between -90°C and +50°C, pressure between 10 and 1200 hPa, it also provide GPS information such as time, longitude, latitude and altitude.'*

*Comments: Line 77: Eq(1):iMET XQ2 should also provide height information, say GPS height,please provide the comparison of the two data.*
*Response: Thanks for the comment. A comparison between GPS height and our calculation is presented. The difference is minimal.*

*Comments: Line 80: please explain "Just go"*
*Response: Thanks for the comments. We want to say this device is easy to set up and portable to be taken for field measurements. We have replaced this word and rewritten the sentences in the revised manuscript.*

*Comments: Line 83: "1300 a.g.l." should be "1300m above ground"??*
*Response: Thanks for the comment. "1300 a.g.l." has been revised to "1300 m above ground level" for consistency and clarity.*

*Comments: Line 87: it's better to change section 2.3 "Lab analysis" to "air sample analysis"*
*Response: Thanks for the comment. Section 2.3 has been renamed from "Lab analysis" to "Air sample analysis" as suggested.*

*Comments: Line 158: CMU or MCU?*
*Response: Thanks for the comment. Yes, it is corrected to "MCU" in the manuscript.*

*Comments: Line 163: what's "GPS profile"?*
*Response: Thanks for the comment. 'GPS profiles' should be corrected by 'potential temperature and relative humidity profiles calculated by iMET XQ2'. However, we have deleted Figure 6-7 and relative texts (Line 163-165) for consistency.*

*The following papers are referred to and added to the list of section References.*

REFERENCES

Britto Hupsel de Azevedo, G., Doyle, B., Fiebrich, C. A., and Schvartzman, D.: Low-complexity methods to mitigate the impact of environmental variables on low-cost UAS-based atmospheric carbon dioxide measurements, Atmos. Meas. Tech., 15, 5599-5618, 10.5194/amt-15-5599-2022, 2022.

Carnell, R. and Senior, C.: Changes in mid-latitude variability due to increasing greenhouse gases and sulphate aerosols, Climate Dynamics, 14, 369-383, 1998.

Friedlingstein, P., O'Sullivan, M., Jones, M. W., Andrew, R. M., Bakker, D. C. E., Hauck, J., Landschützer, P., Le Quéré, C., Luijkx, I. T., Peters, G. P., Peters, W., Pongratz, J., Schwingshackl, C., Sitch, S., Canadell, J. G., Ciais, P., Jackson, R. B., Alin, S. R., Anthoni, P., Barbero, L., Bates, N. R., Becker, M., Bellouin, N., Decharme, B., Bopp, L., Brasika, I. B. M., Cadule, P., Chamberlain, M. A., Chandra, N., Chau, T. T. T., Chevallier, F., Chini, L. P., Cronin, M., Dou, X., Enyo, K., Evans, W., Falk,

S., Feely, R. A., Feng, L., Ford, D. J., Gasser, T., Ghattas, J., Gkritzalis, T., Grassi, G., Gregor, L., Gruber, N., Gürses, Ö., Harris, I., Hefner, M., Heinke, J., Houghton, R. A., Hurtt, G. C., Iida, Y., Ilyina, T., Jacobson, A. R., Jain, A., Jarníková, T., Jersild, A., Jiang, F., Jin, Z., Joos, F., Kato, E., Keeling, R. F., Kennedy, D., Klein Goldewijk, K., Knauer, J., Korsbakken, J. I., Körtzinger, A., Lan, X., Lefèvre, N., Li, H., Liu, J., Liu, Z., Ma, L., Marland, G., Mayot, N., McGuire, P. C., McKinley, G. A., Meyer, G., Morgan, E. J., Munro, D. R., Nakaoka, S. I., Niwa, Y., O'Brien, K. M., Olsen, A., Omar, A. M., Ono, T., Paulsen, M., Pierrot, D., Pocock, K., Poulter, B., Powis, C. M., Rehder, G., Resplandy, L., Robertson, E., Rödenbeck, C., Rosan, T. M., Schwinger, J., Séférian, R., Smallman, T. L., Smith, S. M., Sospedra-Alfonso, R., Sun, Q., Sutton, A. J., Sweeney, C., Takao, S., Tans, P. P., Tian, H., Tilbrook, B., Tsujino, H., Tubiello, F., van der Werf, G. R., van Ooijen, E., Wanninkhof, R., Watanabe, M., Wimart-Rousseau, C., Yang, D., Yang, X., Yuan, W., Yue, X., Zaehle, S., Zeng, J., and Zheng, B.: Global Carbon Budget 2023, Earth Syst. Sci. Data, 15, 5301-5369, 10.5194/essd-15-5301-2023, 2023.

Han, T., Xie, C., Liu, Y., Yang, Y., Zhang, Y., Huang, Y., Gao, X., Zhang, X., Bao, F., and Li, S. M.: Development of a continuous UAV-mounted air sampler and application to the quantification of CO2 and CH4 emissions from a major coking plant, Atmos. Meas. Tech., 17, 677-691, 10.5194/amt-17-677-2024, 2024.

Kunz, M., Lavric, J. V., Gerbig, C., Tans, P., Neff, D., Hummelgård, C., Martin, H., Rödjegård, H., Wrenger, B., and Heimann, M.: COCAP: a carbon dioxide analyser for small unmanned aircraft systems, Atmos. Meas. Tech., 11, 1833-1849, 10.5194/amt-11-1833-2018, 2018.

Liu, Y., Paris, J. D., Vrekoussis, M., Antoniou, P., Constantinides, C., Desservettaz, M., Keleshis, C., Laurent, O., Leonidou, A., Philippon, C., Vouterakos, P., Quéhé, P. Y., Bousquet, P., and Sciare, J.: Improvements of a low-cost CO2 commercial nondispersive near-infrared (NDIR) sensor for unmanned aerial vehicle (UAV) atmospheric mapping applications, Atmos. Meas. Tech., 15, 4431-4442, 10.5194/amt-15-4431-2022, 2022.

Loftfield, N., Flessa, H., Augustin, J., and Beese, F.: Automated gas chromatographic system for rapid analysis of the atmospheric trace gases methane, carbon dioxide, and nitrous oxide, Journal of Environmental Quality, 26, 560-564, 1997.

Masson-Delmotte, V., Zhai, P., Pörtner, H.-O., Roberts, D., Skea, J., Shukla, P. R., Pirani, A., Moufouma-Okia, W., Péan, C., and Pidcock, R.: Global warming of 1.5 C, An IPCC Special Report on the impacts of global warming of, 1, 93-174, 2019.

Ren, W., Tian, H., Xu, X., Liu, M., Lu, C., Chen, G., Melillo, J., Reilly, J., and Liu, J.: Spatial and temporal patterns of CO2 and CH4 fluxes in China's croplands in response to multifactor environmental changes, Tellus B: Chemical and Physical Meteorology, 63, 222-240, 2011.

Reuter, M., Bovensmann, H., Buchwitz, M., Borchardt, J., Krautwurst, S., Gerilowski, K., Lindauer, M., Kubistin, D., and Burrows, J. P.: Development of a small unmanned aircraft system to derive CO2 emissions of anthropogenic point sources, Atmos. Meas. Tech., 14, 153-172, 10.5194/amt-14-153-2021, 2021.

Xie, S.-P., Lu, B., and Xiang, B.: Similar spatial patterns of climate responses to aerosol and greenhouse gas changes, Nature Geoscience, 6, 828-832, 2013.